# No Object Is an Island: Enhancing 3D Semantic Segmentation Generalization with Diffusion Models

**Fan Li** [1]   **Xuan Wang** [1]   **Xuanbin Wang**[1]   **Zhaoxiang Zhang**[1]   **Yuelei Xu**[1]*

[1]Northwestern Polytechnical University
lifan.messages@gmail.com

## Abstract

Enhancing the cross-domain generalization of 3D semantic segmentation is a pivotal task in computer vision that has recently gained increasing attention. Most existing methods, whether using consistency regularization or cross-modal feature fusion, focus solely on individual objects while overlooking implicit semantic dependencies among them, resulting in the loss of useful semantic information. Inspired by the diffusion model's ability to flexibly compose diverse objects into high-quality images across varying domains, we seek to harness its capacity for capturing underlying contextual distributions and spatial arrangements among objects to address the challenging task of cross-domain 3D semantic segmentation. In this paper, we propose a novel cross-modal learning framework based on diffusion models to enhance the generalization of 3D semantic segmentation, named XDiff3D. XDiff3D comprises three key ingredients: (1) constructing object agent queries from diffusion features to aggregate instance semantic information; (2) decoupling fine-grained local details from object agent queries to prevent interference with 3D semantic representation; (3) leveraging object agent queries as an interface to enhance the modeling of object semantic dependencies in 3D representations. Extensive experiments validate the effectiveness of our method, achieving state-of-the-art performance across multiple benchmarks in different task settings. Code is available at https://github.com/FanLiHub/XDiff3D.

## 1   Introduction

3D semantic segmentation, a fundamental task in computer vision with widespread applications in autonomous driving, robotics, and augmented reality, has made significant advancements in recent years [39, 40, 32, 48, 17, 10]. Despite these advancements, it still experiences severe performance degradation when models trained on the source domain are applied to unseen target domains due to the existence of a domain gap. This has sparked growing interest in Domain Generalized 3D Semantic Segmentation (DG3SS) [28, 62, 24, 55, 68], aiming to learn domain-invariant features that enable models to perform well on a variety of unseen target domains with similar semantic distribution.

Current approaches can be broadly categorized into uni-modal methods based solely on point clouds and cross-modal methods that integrate both point cloud and image data, as shown in Figure 1. The former focuses on reducing the domain gap between the source and target domains through domain augmentation [24, 42, 44] or domain mixing [43, 25], but its performance remains limited. The latter exploits image features paired with point clouds to perform consistency regularization [37, 19, 59, 55] or cross-modal feature fusion [54, 28], substantially improving cross-domain generalization and delivering impressive results. However, most existing approaches *treat objects as isolated islands*, focusing solely on domain-invariant features of individual objects and overlooking the implicit

---

*Corresponding author

39th Conference on Neural Information Processing Systems (NeurIPS 2025).

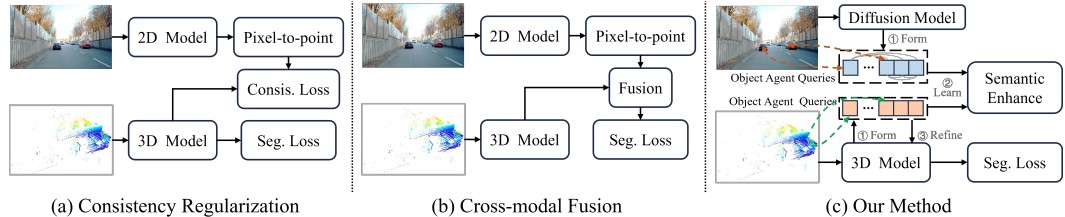

|  (a) Consistency Regularization | (b) Cross-modal Fusion | (c) Our Method |

Figure 1: Existing methods mainly adopt two paradigms: (a) consistency regularization, which enforces alignment across modalities, and (b) cross-modal feature fusion, which integrates multi-modal information. However, both paradigms focus primarily on individual domain-invariant features, overlooking rich semantic dependencies among objects. In contrast, our method (c) leverages object agent queries as an interface to incorporate instance semantic dependencies from diffusion priors into the 3D semantic space, enhancing cross-domain generalization of 3D semantic segmentation.

semantic dependencies among them such as spatial arrangements and contextual relations, which leads to the loss of informative cues and ultimately leads to suboptimal performance.

Recently, diffusion models have provided a new perspective for improving semantic segmentation generalization with remarkable capabilities in capturing underlying semantic relations among objects to synthesize high-quality samples across diverse domains [18, 12, 41]. Building on this insight, several studies [38, 22, 35, 52] have leveraged diffusion priors to better model semantic dependencies among objects, thereby enhancing the generalization of 2D semantic segmentation. Given the remarkable progress of diffusion models in the 2D visual domain, a natural curiosity has been raised: **How can the prior knowledge encoded in diffusion models be leveraged to improve the generalization of 3D semantic segmentation?**

A naive solution is to treat diffusion models as general feature extractors and utilize their features as supervisory signals or auxiliary information by pixel-to-point matching for 3D semantic segmentation. However, this comes with two new challenges: (1) *Strict pixel-to-point matching leads to significant loss of image features due to point cloud sparsity* [1, 37], *further aggravated by mainstream diffusion models operating in compressed latent spaces rather than pixel spaces, hindering effective cross-modal feature association.* (2) *Diffusion models are inherently designed for generative tasks, and their feature spaces not only model high-level instance semantics but also retain fine-grained local visual details (e.g., slogans and scrawl) that may introduce noise and disturb 3D feature learning.*

To address the above challenges, we propose XDiff3D, a cross-modal learning framework via diffusion models for cross-domain 3D semantic segmentation, which leverages the semantic dependencies among objects embedded in pretrained diffusion priors to enrich the learned representations of point clouds. Specifically, for **the first challenge**, we introduce learnable queries that interact with diffusion features to aggregate rich semantic dependencies among objects, forming object agent queries that subsequently refine the 3D feature representations. To tackle **the second challenge**, we introduce a dual-query learning scheme that enforces semantic consistency between the principal components of two sets of queries, effectively suppressing interference from potentially distracting visual details. Ultimately, these object agent queries serve as an interface to infuse 2D semantic priors into the 3D representation space, enabling the refinement of 3D semantic features in the decoder for more generalizable representations. Extensive experiments across multiple benchmark settings demonstrate that XDiff3D consistently delivers state-of-the-art performance, surpassing existing baselines by a significant margin. Moreover, comprehensive ablation studies and analyses further validate the effectiveness of its core components. The contributions are summarized as follows:

- To the best of our knowledge, XDiff3D is the first diffusion-based cross-modal learning framework that constructs object agent queries as an interface to infuse 2D diffusion priors into the 3D representation space, thereby enhancing the cross-domain generalization of 3D semantic segmentation.

- We propose a dual-queries learning scheme that suppresses potentially intricate visual details embedded in diffusion priors to prevent interference with the learning of robust 3D feature representations.

- XDiff3D is a concise and general framework, consistently outperforming a wide variety of baselines and achieving state-of-the-art performance across multiple benchmarks for cross-domain 3D semantic segmentation.

## 2 Related Works

### 2.1 Diffusion Models

Diffusion models [18, 46, 12, 41] have demonstrated impressive capabilities in image generation and are increasingly being explored for visual perception tasks such as semantic segmentation [60, 53, 49], object detection [7, 20], and depth estimation [23]. Recent studies have begun to exploit diffusion models for improving domain generalization in 2D segmentation [38, 35, 4]. For example, DGInStyle [21] introduces a controllable framework that generates diverse, task-specific images from diffusion priors. Niemeijer et al. [35] use text-guided diffusion to synthesize pseudo-target domains for better coverage of target domain variations. Despite the progress in 2D tasks, the potential of diffusion priors to enhance generalization in 3D semantic segmentation remains underexplored. This work addresses this gap by leveraging instance-level semantic priors from diffusion models to enrich 3D semantic representations and improve their generalization with respect to domain shifts.

### 2.2 Domain Generalized 3D Semantic Segmentation

Domain Generalized 3D Semantic Segmentation (DG3SS) aims to learn a model solely from source domain data that can perform well across diverse unseen target domains, and has recently garnered significant attention in the research community [5, 62, 58, 55, 68]. SemanticSTF [58] introduces a large-scale benchmark for semantic segmentation of LiDAR point clouds in adverse weather, enabling comprehensive evaluation of domain adaptive and generalizable 3D segmentation methods under all-weather conditions. MM2D3D [5] injects depth cues into the 2D branch and RGB information into the 3D branch to improve modality complementarity and robustness to domain shift. 2DPASS [62] introduces a fusion-then-distillation strategy to transfer rich semantic and structural information from 2D images to 3D point clouds without requiring strictly paired data. UniDSeg [55] leverages Visual Foundation Models by introducing learnable prompts within a cross-modal framework to bridge the 2D-3D domain gap and enhance generalization in cross-domain 3D semantic segmentation. Despite these advancements, existing methods primarily focus on learning domain-invariant features for individual objects, while overlooking the latent semantic dependencies among objects. In contrast, this paper exploits the contextual relationships and spatial organization among objects to move beyond isolated instance modeling, enabling the model to grasp a unified semantic distribution underlying diverse scenes, thereby enhancing cross-domain generalization.

### 2.3 Learnable Queries Design

Recently, various approaches [36, 30, 66, 51] have adopted learnable query-based frameworks inspired by DETR [6]. MaskFormer [8] and Mask2Former [9] unify semantic and instance segmentation using object queries. Tqdm [36] introduces domain-invariant textual queries for domain generalized semantic segmentation. kMaX-DeepLab [64] treats pixel–query interaction as k-means-style clustering, simplifying cross-attention for better segmentation. While learnable queries have proven effective in various perception tasks, how to leverage them to enhance generalization in 3D semantic segmentation remains an open question. In this work, we derive object agent queries from diffusion features and utilize them as a cross-modal interface to enhance 3D semantic representations.

## 3 Method

### 3.1 Preliminary

**Stable Diffusion.** Stable diffusion models comprise two complementary stochastic processes: a diffusion process and a reverse process. During the diffusion process, random noise is progressively added to the data via a Markov chain:

$$q\left(z_t \mid z_0\right) := \mathcal{N}\left(z_t \mid \sqrt{\bar{\alpha}_t} z_0, \left(1 - \bar{\alpha}_t\right) I\right), \quad z_0 = \mathcal{E}(x) \tag{1}$$

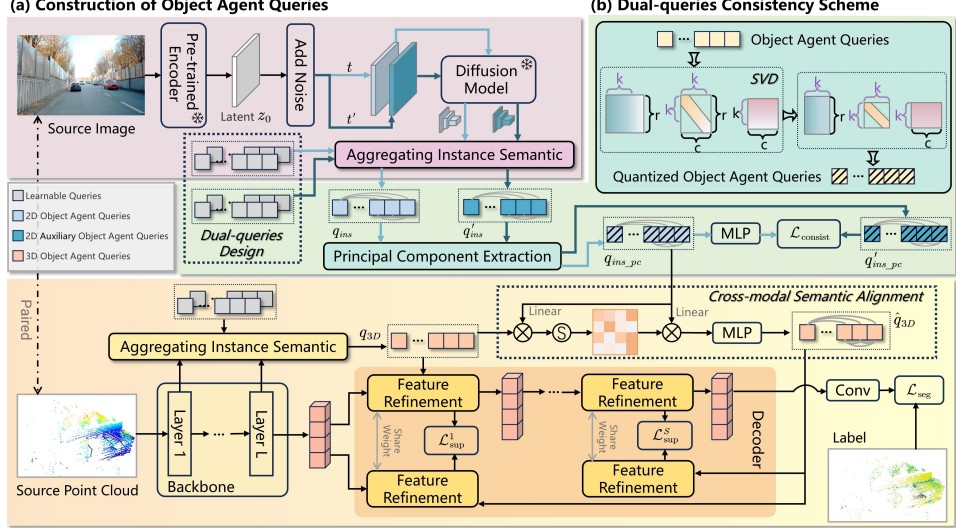

Figure 2: A brief illustration of our proposed framework. First, we aggregate semantic information of objects within the scene from diffusion features to form the object agent queries. Next, we propose a dual-query mechanism to eliminate local visual details from the object agent queries, preventing interference with 3D semantic representation. Finally, the optimized object agent queries are used as an interface to infuse inter-object semantic dependencies into the 3D representation, guiding the model to learn domain-invariant features.

where $\mathcal{E}$ denotes a pretrained VAE encoder that maps an input image $x \in \mathbb{R}^{H \times W \times 3}$ into a latent representation $z_0$. The hyperparameters $\bar{\alpha}_t$ represent a pre-defined noise schedule, where a larger $t$ corresponds to larger noise weights. The reverse process gradually reconstructs clean data from noisy samples using a noise predictor $\epsilon_\theta(\cdot)$. Each step of this reverse process can be formulated as:

$$p_\theta\left(z_{t-1} \mid z_t\right) := \mathcal{N}\left(z_{t-1} \mid \mu_\theta\left(z_t, t\right), \Sigma_\theta\left(z_t, t\right)\right) \tag{2}$$

where $\mu_\theta$ is the mean predicted by $\epsilon_\theta(\cdot)$, and $\Sigma_\theta$ is typically set to a predefined covariance value.

**Problem Definition.** The goal of DG3SS is to train a segmentation model solely on a labeled source domain $\mathcal{S}$, enabling it to generalize to unseen target domains $\mathcal{T}$. For the source domain, it is assumed that paired images and point clouds are available. In general, the segmentation model $\varphi = g \circ \upsilon$ consists of a backbone $g$ for feature extraction and a decoder $\upsilon$ for producing semantic predictions.

## 3.2 Overview

With diffusion models successfully enhancing segmentation generalization in the 2D vision domain drawing on strong instance semantic priors, a natural question arises: How can the prior knowledge encoded in diffusion models be leveraged to bolster the generalization of 3D semantic segmentation? As illustrated in Figure 2, we propose XDiff3D, a cross-modal learning framework for cross-domain 3D semantic segmentation based on diffusion models, which consists of three key ingredients: (1) aggregating object semantic information from the prior knowledge encoded in diffusion features to generate object agent queries (Section 3.3), (2) suppressing potential intricate visual details embedded within the object agent queries (Section 3.4), and (3) leveraging the optimized object agent queries as an interface to perform cross-modal semantic enhancement (Section 3.5).

## 3.3 Construction of Object Agent Queries

Given that diffusion models operate in compressed latent spaces, traditional calibration-based hard associations between LiDAR points and image pixels become unreliable. To fully harness the rich instance semantic priors captured by diffusion features, we construct object agent queries from the diffusion space and employ them as an interface for infusing semantic knowledge into the 3D feature space, with the goal of enhancing the generalization capability of 3D semantic segmentation.

**Diffusion Feature Extraction.** Before that, we need to extract diffusion features. To do this, we begin by encoding the source image $x$ into a latent code $z_0 = \mathcal{E}(x)$ via a pretrained encoder $\mathcal{E}$. Subsequently, we introduce noise to $z_0$ according to Equation 1, yielding the noisy latent code $z_t$. Finally, we apply the noise predictor $\epsilon_\theta(\cdot)$ to perform the denoising process to get diffusion features:

$$f_{sd}^{(i)} = \epsilon_\theta^i\left(z_t, t, \mathcal{T}(e)\right), \text{with } 1 \leq i \leq M \tag{3}$$

where $\mathcal{T}$ denotes the text encoder, $f_{sd}^{(i)}$ represents the $i$-th layer features from the diffusion model, $e$ is the empty text, and $M$ is the total number of diffusion feature layers.

**Aggregating Instance Semantic Features.** We employ multiple sets of learnable queries, each corresponding to a specific layer of diffusion features. Specifically, for the $i$-th layer of diffusion features, we map them to key ($\mathbf{K}^i$) and value ($\mathbf{V}^i$) vectors, while a corresponding set of randomly initialized learnable queries is mapped to query ($\mathbf{Q}^i$) vectors:

$$\mathbf{Q}^i = q_{init}^i \mathbf{W}_{\mathcal{Q}}^i, \ \mathbf{K}^i = f_{sd}^i \mathbf{W}_{\mathcal{K}}^i, \ \mathbf{V}^i = f_{sd}^i \mathbf{W}_{\mathcal{V}}^i, \ \text{with } q_{init}^i \in \mathbb{R}^{r \times c} \tag{4}$$

where $\mathbf{W}_{\mathcal{Q}}^i$, $\mathbf{W}_{\mathcal{K}}^i$ and $\mathbf{W}_{\mathcal{V}}^i$ are linear projection matrices, $q_{init}^i$ denotes randomly initialized learnable queries, $c$ is the dimension of $q_{init}^i$, and $r$ is the sequence length of $q_{init}^i$. Next, the layer-wise object agent queries are formed as follows:

$$\hat{q}_{ins}^i = \text{FFN}\left(\frac{\exp\left(s^i\right)}{\sum_{j=1}^{hw} \exp\left(s^i\right)} \times \mathbf{V}^i\right), \ s^i = \frac{\mathbf{Q}^i(\mathbf{K}^i)^T}{\sqrt{d^i}}, \ \text{with } \hat{q}_{ins}^i \in \mathbb{R}^{r \times c} \tag{5}$$

where $\hat{q}_{ins}^i$ are the object agent queries of the $i$-th diffusion layer, $d^i$ is a scaling factor, and FFN consists of a linear mapping followed by an activation layer. $h$ and $w$ denote the height and width of the similarity matrix $s^i$, respectively. After the final layer, $M$, we compute both the maximum and average components across all layer-wise object agent queries to obtain the global object agent queries:

$$q_{ins} = \left(\max_{i=1,2,\ldots,M} \hat{q}_{ins}^i + \frac{1}{M}\sum_{i=1}^{M} \hat{q}_{ins}^i\right) \times W_a + b_a, \ \text{with } q_{ins} \in \mathbb{R}^{r \times c} \tag{6}$$

where $W_a$ and $b_a$ signify the weights and biases, respectively. Average queries encode global contextual information to improve robustness to noise, while max queries emphasize the most prominent and distinctive signals, highlighting key semantic features. As a result, the object agent queries establish implicit associations with scene instances and learn the semantic dependencies among objects embedded in diffusion features. Acting as an interface, these queries guide the 3D feature representations to move beyond isolated individuals, enabling the model to grasp consistent semantic patterns across diverse scenario domains.

## 3.4 Dual-queries Consistency Scheme

As diffusion models are inherently designed for generative tasks, their feature spaces encode not only high-level instance semantics but also intricate local visual details. These details, though irrelevant to 3D semantic segmentation, may be inadvertently propagated into object agent queries during interaction process and disrupt the learning of 3D semantic representations. To this end, we propose a dual-query consistency scheme to decouple potential local visual details from the object agent queries.

**Dual-queries Design.** To achieve this, we first construct an auxiliary set of object agent queries $q_{ins}'$ guided by diffusion features at higher timesteps $t'$, following the procedure described in Section 3.3. Different timesteps in the diffusion process correspond to successive stages of denoising. Higher timesteps introduce stronger noise that blurs local visual details, while the overall semantic structure of the scene remains largely preserved. As a result, the primary difference between the diffusion features at timestep $t'$ and $t$ lies in the superficial visual details.

**Principal Component Extraction and Consistency Constraint.** Recently, SoMA [65] reveals that the principal components derived from singular value decomposition (SVD) of weight matrices in vision foundation models (VFMs) capture generalized world knowledge, which underpins their strong generalization capability. Inspired by this, the object agent queries, as learnable parameters, are expected to exhibit similar properties with principal components that emphasize domain-invariant

instance semantic knowledge rather than intricate visual details. To this end, we first perform singular value decomposition (SVD) on the two sets of object agent queries:

$$q_{ins} = U\Sigma\mathcal{V}^T, \quad q'_{ins} = U'\Sigma'\mathcal{V}^{T,'}, \quad \text{with } U \in \mathbb{R}^{r \times r}, \Sigma \in \mathbb{R}^{r \times c}, \mathcal{V} \in \mathbb{R}^{c \times c} \tag{7}$$

where $U$ (or $U'$) and $\mathcal{V}$ (or $\mathcal{V}'$) are the left and right singular vectors, $\Sigma$ (or $\Sigma'$) is a diagonal matrix whose entries are singular values arranged in descending order. We select the top-$k$ singular values along with their associated components in $U$ (or $U'$) and $\mathcal{V}$ (or $\mathcal{V}'$) to generate the quantized object agent queries $q_{ins\_pc}$ and $q'_{ins\_pc}$:

$$q_{ins\_pc} = U_{[:,:k]}\Sigma_{[:k]}\mathcal{V}^T_{[:k,:]}, \; q'_{ins\_pc} = U'_{[:,:k]}\Sigma'_{[:k]}\mathcal{V}^{T,'}_{[:k,:]}, \text{ with } q_{ins\_pc} \in \mathbb{R}^{r \times c}, q'_{ins\_pc} \in \mathbb{R}^{r \times c} \tag{8}$$

Subsequently, we impose a consistency constraint between the two sets of quantized agent queries:

$$\mathcal{L}_{\text{consist}} = \sum_{\hat{r}=1}^{r} q'_{ins\_pc}(\hat{r}) \log \frac{q'_{ins\_pc}(\hat{r})}{\text{MLP}(q_{ins\_pc})(\hat{r})} \tag{9}$$

where the MLP includes a linear transformation followed by layer normalization. By enforcing a consistency constraint, the principal components of the object agent queries are effectively guided to focus on instance semantic distribution rather than local visual details. This is because encoding excessive visual details within the principal components of the object agent queries would result in substantial discrepancies between the two sets of queries, $q_{ins\_pc}$ and $q'_{ins\_pc}$. Thus, the consistency loss reduces these discrepancies, ensuring that only semantic dependencies among objects are retained in the constructed object agent queries, while irrelevant visual details are effectively suppressed.

### 3.5 Cross-modal Semantic Enhancement

Our ultimate goal is to improve the cross-domain generalization of 3D segmentation models. To this end, we leverage object agent queries as an interface for cross-modal semantic alignment, which subsequently enables the refinement of 3D feature representations to better capture semantic dependencies among objects across domains.

**Cross-modal Semantic Alignment.** Before that, we first generate the 3D object agent queries corresponding to the point cloud features. Similarly, we introduce multiple sets of learnable queries, each corresponding to a specific layer of the 3D backbone. These queries are projected into query vectors, while the associated 3D features are transformed into key and value vectors. A cross-attention mechanism is then employed to facilitate interaction between the learnable queries and 3D features, yielding layer-specific 3D agent queries. To generate the final 3D object agent query, we aggregate the outputs from all layers using both max and average pooling, followed by an MLP projection. For clarity, the previously introduced object agent queries derived from diffusion features are referred to as 2D object agent queries in the remainder of this paper. Next, we perform a dot product operation on the 2D object agent queries $q_{ins\_pc}$ with the 3D object agent queries $q_{3D}$ to obtain a similarity map:

$$S = \text{Softmax}\left(q_{3D} \times \text{Linear}(q_{ins\_pc})^T\right), \quad \text{with } S \in \mathbb{R}^{r \times r} \tag{10}$$

where Linear is a two-layer MLP with layer normalization. The dot product operation produces a similarity matrix $S$, explicitly linking instance semantic information between the 2D and 3D object agent queries. Subsequently, we leverage $S$ to recompose the semantic information from the 2D object agent queries, resulting in a new set of enhanced 3D object agent queries $\hat{q}_{3D}$ infused with the object-wise semantic dependencies conveyed by the 2D queries:

$$\hat{q}_{3D} = \text{MLP}(S \times q_{ins\_pc} + q_{3D}), \quad \text{with } \hat{q}_{3D} \in \mathbb{R}^{r \times c} \tag{11}$$

**Feature Refinement and Mask Prediction.** The enhanced 3D object agent queries $\hat{q}_{3D}$ act as supervisory signals to guide the original 3D agent queries in actively modeling inter-instance semantic dependencies during their construction process. Inspired by UniDSeg [55] and xMUDA [19], we refine the segmentation features in the 3D decoder using $\hat{q}_{3D}$, while encouraging $q_{3D}$ to mimic this refinement process:

$$\hat{f}^s_{3D\_aug} = \text{MLP}(\text{Softmax}(f^s_{3D} \times (\hat{q}_{3D})^T) \times \hat{q}_{3D}), \; \hat{f}^s_{3D} = \text{MLP}(\text{Softmax}(f^s_{3D} \times (q_{3D})^T) \times q_{3D})$$

$$\mathcal{L}^s_{\text{sup}} = \sum_{j=1}^{K} L_\delta(\hat{f}^s_{3D}, \hat{f}^s_{3D\_aug}), \quad \text{with } L_\delta(x,y) = \begin{cases} 0.5(x-y)^2 & \text{if } |x-y| < 1 \\ |x-y| - 0.5 & \text{otherwise} \end{cases}$$

$$\tag{12}$$

Table 1: Performance comparison of domain adaptive and domain generalized 3D semantic segmentation methods in four typical settings. Top three results are highlighted as best , second and third , respectively. xM denotes the result which is obtained by taking the mean of the predicted 2D and 3D probabilities after softmax.

| S:Source /T:Target | | vKITTI/sKITTI | | nuScenes:USA/Sing | | nuScenes:Day/Night | | A2D2/sKITT | |
|---|---|---|---|---|---|---|---|---|---|
| Task | Method | 3D | xM | 3D | xM | 3D | xM | 3D | xM |
| DA | logCORAL [34] | 36.8 | 47.0 | 63.2 | 69.4 | 68.7 | 63.7 | 41.0 | 42.2 |
| | MinEnt [50] | 43.3 | 47.1 | 61.5 | 66.0 | 68.8 | 63.6 | 39.6 | 42.6 |
| | BDL [29] | 44.3 | 35.6 | 64.8 | 70.4 | 69.6 | 63.0 | 41.7 | 45.2 |
| | xMUDA [19] | 46.7 | 48.2 | 63.2 | 69.4 | 69.2 | 67.4 | 46.0 | 44.0 |
| | AUDA [31] | 37.8 | 41.3 | 64.0 | 69.2 | 69.8 | 64.8 | 43.6 | 46.8 |
| | DsCML [37] | 38.4 | 45.5 | 56.2 | 66.1 | 49.3 | 53.2 | 45.1 | 44.5 |
| | Dual-Cross [27] | 35.1 | 44.2 | 58.1 | 66.5 | 69.7 | 68.0 | 40.0 | 48.6 |
| | SSE [67] | 40.0 | 49.6 | 63.9 | 69.2 | 69.0 | 68.9 | 46.8 | 48.4 |
| | BFtD [54] | 45.5 | 51.5 | 62.2 | 69.4 | 70.4 | 68.3 | 44.4 | 48.7 |
| | MM2D3D [5] | 50.3 | 56.5 | 66.8 | 72.4 | 70.2 | 72.1 | 46.1 | 46.2 |
| | VFMSeg [61] | 52.0 | 61.0 | 65.6 | 72.3 | 70.5 | 66.5 | 52.3 | 50.0 |
| | UniDSeg [55] | 50.9 | 62.0 | 67.6 | 72.9 | 71.2 | 71.2 | 55.4 | 57.5 |
| | **XDiff3D** | **53.1** | **63.3** | **69.5** | **74.1** | **73.5** | **72.3** | **57.6** | **58.8** |
| DG | xMUDA [19] | 37.4 | 39.0 | 62.3 | 68.6 | 68.9 | 59.6 | 36.7 | 41.6 |
| | MM2D3D [5] | 40.2 | 44.2 | 62.3 | 70.9 | 63.2 | 68.3 | 35.9 | 43.6 |
| | UniDSeg [55] | 44.7 | 60.0 | 64.5 | 72.3 | 70.5 | 70.0 | 46.3 | 54.4 |
| | **XDiff3D** | **46.9** | **61.3** | **66.7** | **73.5** | **72.4** | **71.6** | **49.1** | **56.2** |

where $f_{3D}^s$ denotes $s$-th stage features of the 3D decoder and $1 \leq s \leq S$. After the final stage, $S$, the refined segmentation features $\hat{f}_{3D}^S$ are fed into a 3D convolution to generate the final segmentation predictions:

$$\mathcal{L}_{\text{seg}} = \mathcal{L}_{\text{ce}}\left(\text{Conv}(\hat{f}_{3D}^S), Y\right) \tag{13}$$

where $Y$ denotes the point cloud labels and $\mathcal{L}_{\text{ce}}$ denotes the cross-entropy loss.

**Full Objective.** Ultimately, the overall objective of the training process is defined as follows:

$$\mathcal{L}_{total} = \mathcal{L}_{\text{seg}} + \gamma \mathcal{L}_{\text{sup}} + \lambda \mathcal{L}_{\text{consist}} \tag{14}$$

where $\gamma$ and $\lambda$ are hyperparameters. Notably, all operations involving 2D object agent queries and diffusion models are discarded during inference to ensure our framework remains concise and general.

## 4 Experiments

### 4.1 Datasets and Metrics

Following prior works [19, 58, 55, 68], we evaluate our method on six publicly available autonomous driving datasets, comprising three real-world datasets (nuScenes [13], SemanticKITTI [3], and A2D2 [15]), two synthetic datasets (VirtualKITTI [14] and SynLiDAR [57]), and one adverse-weather dataset (SemanticSTF [58]). The real-world datasets provide synchronized and calibrated LiDAR and RGB sensor data, enabling direct 2D-to-3D projection, whereas VirtualKITTI includes depth maps from which we simulate LiDAR scans by uniformly sampling points. For more details about the datasets, please refer to the supplementary material. Following standard practice [58, 55, 68], we evaluate segmentation performance using the mean Intersection over Union (mIoU) averaged over all classes for each dataset.

### 4.2 Implementation Details

Following prior works [55, 19], we adopt SparseConvNet [16] with a U-Net-style architecture as our 3D segmentation model, implemented using the Sparse Convolution Library [11]. The voxel resolution is set to 5cm, ensuring that each voxel encapsulates a single 3D point and provides sufficient spatial granularity for semantic segmentation. For the diffusion model, we leverage Stable Diffusion v2-1 [41], pretrained on the LAION-5B dataset [45], which is kept frozen during the entire training process. The model is optimized using AdamW [33], with a learning rate of 1e-5 for the 3D backbone

Table 2: Comparison of previous domain generalization methods on SemanticKITTI→SemanticSTF and SynLiDAR→SemanticSTF benchmarks.

| Method | Dense-fog | Light-fog | Rain | Snow | Dense-fog | Light-fog | Rain | Snow |
|---|---|---|---|---|---|---|---|---|
| | SemanticKITTI→SemanticSTF | | | | SynLiDAR→SemanticSTF | | | |
| Dropout [47] | 29.3 | 25.6 | 29.4 | 24.8 | 15.3 | 16.6 | 20.4 | 14.0 |
| Perturbation [58] | 26.3 | 27.8 | 30.0 | 24.5 | 16.3 | 16.7 | 19.3 | 13.4 |
| PolarMix [56] | 29.7 | 25.0 | 28.6 | 25.6 | 16.1 | 15.5 | 19.2 | 15.6 |
| MMD [26] | 30.4 | 28.1 | 32.8 | 25.2 | 17.3 | 16.3 | 20.0 | 12.7 |
| PCL [63] | 28.9 | 27.6 | 30.1 | 24.6 | 17.8 | 16.7 | 19.3 | 14.1 |
| PointDR [58] | 31.3 | 29.7 | 31.9 | 26.2 | 19.5 | 19.9 | 21.1 | 16.9 |
| UniMix [68] | 34.8 | 30.2 | 34.9 | 30.9 | 24.3 | 22.9 | 26.1 | 20.9 |
| **XDiff3D** | **37.5** | **34.1** | **38.1** | **33.7** | **26.2** | **24.6** | **27.9** | **22.5** |

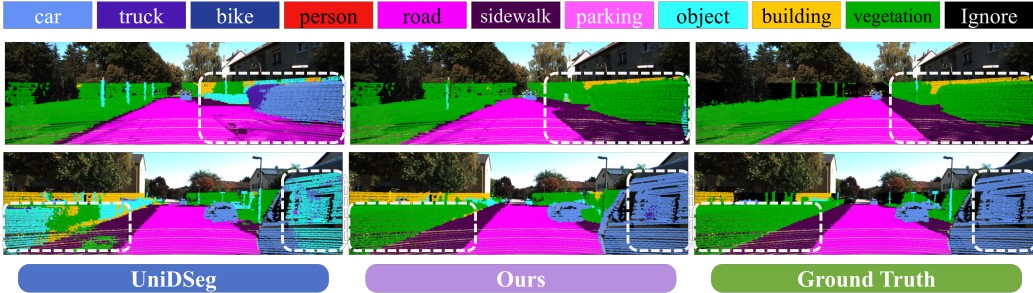

Figure 3: Qualitative results of DG3SS. From left to right: the visual results predicted by UniDSeg, Ours, and Ground Truth. We deploy the white dash boxes to highlight different prediction parts.

and 1e-4 for the decoder and learnable queries. Training is conducted for 50000 iterations with a batch size of 8. All experiments are performed on 4 NVIDIA RTX 4090 GPUs.

## 4.3 Comparison with State-of-the-art Methods

We comprehensively compare our method with existing domain adaptive 3D semantic segmentation (DA3SS) and domain generalized 3D semantic segmentation (DG3SS) methods. We conduct experiments in three standard evaluation settings: synthetic-to-real (VirtualKITTI→SemanticKITTI), real-to-real (nuScenes:USA→Sing, nuScenes:Day→Night, and A2D2→SemanticKITTI) and normal-to-adverse (SemanticKITTI→SemanticSTF and SynLiDAR→SemanticSTF). For the 2D branch architecture, we adopt the same structure as UniDSeg [55]. For DA3SS experiments, we also follow the exact configurations and training protocols of UniDSeg to ensure consistency and fair comparison.

**Synthetic-to-real generalization.** In Table 1, we compare our method with existing DG3SS and DA3SS approaches under the VirtualKITTI→SemanticKITTI setting. Our method significantly outperforms the previous state-of-the-art method UniDSeg in both 3D-only and 2D–3D fusion (i.e. xM) settings. Notably, in the 3D-only configuration, our method surpasses UniDSeg by 2.2 points in mIoU, demonstrating its strong cross-domain generalization capability.

**Real-to-real generalization.** In this experimental setting, models are evaluated under the real-to-real benchmarks, including nuScenes:USA→Sing, nuScenes:Day→Night, and A2D2→SemanticKITTI. As illustrated from the fifth column onward in Table 1, our method consistently achieves superior performance across all datasets in both DG and DA scenarios, surpassing previous state-of-the-art methods by clear margins. These results highlight the robustness and strong generalization capability of our framework in effectively handling complex and realistic domain variations encountered in practical 3D semantic segmentation tasks. Figure 3 further provides qualitative comparisons on the A2D2→SemanticKITTI benchmark, where our approach yields more complete and coherent predictions for road, sidewalk, building, and vegetation regions, effectively resolving ambiguous boundaries and reducing unreasonable predictions observed in UniDSeg. More qualitative results and detailed analyses are included in the supplementary material.

**Normal-to-adverse generalization.** In this experimental setting, all models are evaluated on the SemanticSTF dataset, which includes a range of challenging weather conditions. As shown in Table

Table 3: Ablation study on primary components, where 3D_AQ denotes 3D object agent queries.

| | 3D_AQ | $\mathcal{L}_{sup}$ | $\mathcal{L}_{consist}$ | nuScenes:USA/Sing | | nuScenes:Day/Night | | A2D2/sKITT | |
|---|---|---|---|---|---|---|---|---|---|
| | | | | 3D | xM | 3D | xM | 3D | xM |
| 1 | | | | 64.5 | 72.3 | 70.5 | 70.0 | 46.3 | 54.4 |
| 2 | ✓ | | | 64.9 | 72.5 | 70.7 | 70.1 | 46.8 | 54.7 |
| 3 | ✓ | ✓ | | 66.0 | 73.1 | 71.6 | 71.2 | 48.0 | 55.6 |
| 4 | ✓ | ✓ | ✓ | 66.7 | 73.5 | 72.4 | 71.6 | 49.1 | 56.2 |

Table 4: Ablation study on different timestep selections under the DG3SS setting.

| $t$ | $t'$ | nuScenes:USA/Sing | | A2D2/sKITT | |
|---|---|---|---|---|---|
| | | 3D | xM | 3D | xM |
| 0 | 50 | 66.1 | 72.9 | 48.4 | 55.6 |
| 50 | 75 | 66.5 | 73.4 | 48.6 | 56.0 |
| 50 | 150 | **66.7** | **73.5** | **49.1** | **56.2** |
| 150 | 300 | 66.3 | 73.1 | 48.3 | 55.6 |
| 300 | 400 | 65.7 | 72.7 | 47.8 | 55.3 |

Table 5: Comparison of diffusion models under DG3SS on VirtualKITTI→SemanticKITTI.

| Model | UniDSeg | +SD 1.4 | +SD 1.5 | +SD 2.1 |
|---|---|---|---|---|
| mIoU | 44.7 | **46.1** | **46.6** | **46.9** |

Table 6: Ablation on principal components under DG3SS on VirtualKITTI→SemanticKITTI.

| $k$ | 20 | 50 | 70 | 90 |
|---|---|---|---|---|
| mIoU | 45.7 | 46.1 | **46.9** | 46.7 |

2, our method consistently outperforms existing approaches across all adverse scenarios. Notably, under the SemanticKITTI→SemanticSTF setting, our approach achieves a significant improvement over the previous state-of-the-art, exceeding it by more than 2 mIoU points on average. These results demonstrate the strong robustness and generalization ability of our method in the face of severe domain shifts.

**Comparison of the class-wise IoU.** We present a class-wise IoU analysis in Figure 4, using UniDSeg as the baseline model. The results reveal consistent improvements across a wide range of categories. The heatmap further illustrates the robustness of our method in capturing meaningful semantic structures and maintaining performance across domain shifts.

## 4.4 Ablation Study

This section presents comprehensive experimental results to verify the effectiveness of the proposed method. For more ablation studies and detailed analyses, please refer to the supplementary material.

**Effect of components.** We conduct ablation studies under DG3SS settings to validate the effectiveness of key components in our proposed method, specifically examining the contributions of the 3D object agent queries, $\mathcal{L}_{sup}$ and $\mathcal{L}_{consist}$. Using UniDSeg as our baseline model, the results presented in Table 3 reveal that: (1) Each component individually enhances performance, confirming their respective effectiveness. (2) Integrating all three components yields the highest mIoU scores across all benchmarks, underscoring their complementary roles in improving cross-domain segmentation generalization.

**Study of the object agent query dimension $c$.** The object agent queries serve as the core component of our framework. To assess the impact of their feature dimensionality, we experiment with values ranging from 64 to 1024. As shown in Figure 5, setting $c = 256$ yields competitive performance, achieving mIoU scores of 66.7% and 72.4% on the nuScenes:USA/Sing and nuScenes:Day/Night benchmarks, respectively.

**The choice of $t$ and $t'$.** Different timesteps in the diffusion process correspond to different denoising stages, with larger timesteps introducing stronger noise. Prior studies [60, 2] indicate that effective timesteps typically fall within the range of 0 to 300, and that adjacent timesteps often yield highly similar features. To ensure sufficient diversity while maintaining semantic integrity, we select representative intervals from this range. As shown in Table 4, we empirically choose $t = 50$ and $t' = 150$ to effectively capture high-level semantic information while minimizing interference from low-level visual details.

**Comparing different stable diffusion.** As illustrated in Table 5, we adopt UniDSeg as the baseline and compare the performance of our method using three representative stable diffusion models. The results demonstrate that our method is robust to the choice of diffusion model, consistently outperforming the baseline and achieving significant performance gains.

Table 7: Effect of using different 3D backbones on the DA3SS methods.

| 3D Backbone | DA3SS | USA/Sing | |
|---|---|---|---|
| | | 3D | xM |
| SparseConvNet | UniDSeg | 67.6 | 72.9 |
| | XDiff3D | 69.5 | 74.1 |
| MinkowskiNet | UniDSeg | 68.6 | 73.1 |
| | XDiff3D | 70.2 | 74.3 |

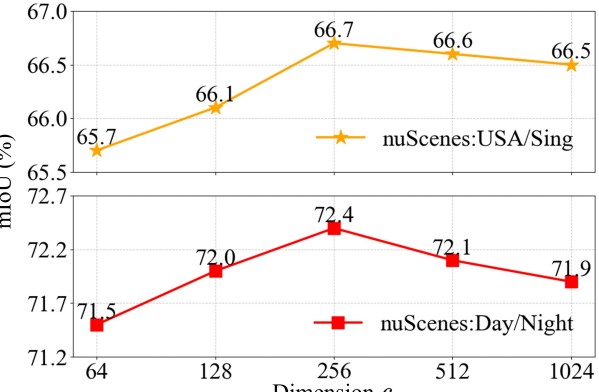

| | Veget. | Build. | Road | Object | Truck | Car |
|---|---|---|---|---|---|---|
| Base | 60.66 | 53.37 | 82.24 | 4.64 | 24.53 | 80.13 |
| Ours | 61.18 | 58.04 | 83.26 | 5.09 | 30.02 | 81.64 |

Figure 4: Comparison of the class-wise IoU on VirtualKITTI → SemanticKITTI under the DA3SS setting, using UniDSeg with and without our method.

Figure 5: Ablation study on agent queries dimension $c$.

**The number of principal components $k$.** The results in Table 6 indicate that the selection of $k$—the number of retained principal components—significantly influences segmentation performance, with $k = 70$ achieving the highest mIoU. Increasing $k$ beyond this value introduces fine-grained visual details, which interferes with the learning of domain-invariant semantics and leads to performance degradation.

**Ablation on different backbones.** As shown in Table 7, we evaluate the performance of various backbones integrated into our framework under consistent parameter settings. The results indicate that stronger backbones yield improved performance, and our method consistently outperforms the baseline across all configurations.

Table 8: Performance comparison under different noise levels.

| $\sigma$ (m) | 0.05 | 0.1 | 0.2 | 0.3 |
|---|---|---|---|---|
| UniDSeg (baseline) | 40.3 | 37.1 | 33.8 | 30.6 |
| Ours | **44.7** | **43.2** | **39.6** | **34.7** |

**Robustness to noisy point clouds.** To evaluate robustness against measurement noise, we conducted an ablation study on A2D2 → SemanticKITTI benchmarks by adding zero-mean Gaussian noise with varying standard deviations $\sigma$. As shown in Table 8, our method consistently outperforms the UniDSeg baseline across all noise levels, maintaining higher mIoU even under severe perturbations. This demonstrates that our model possesses strong robustness and effectively preserves semantic consistency under noisy conditions.

## 5 Conclusion

In this work, we propose XDiff3D, a novel cross-modal framework guided by diffusion models to enhance the generalization of 3D semantic segmentation. XDiff3D constructs object agent queries to capture semantic dependencies among objects from diffusion features and infuses them into the 3D representation space. To mitigate interference from visual details, we further propose a dual-queries consistency scheme that encourages object-agent queries to focus on domain-invariant semantics. Extensive experiments on DG3SS and DA3SS benchmarks demonstrate that XDiff3D significantly outperforms previous SOTA methods, underscoring its effectiveness. This work addresses a significant gap in existing research and sets a new benchmark for cross-domain 3D semantic segmentation.

## Acknowledgments and Disclosure of Funding

This work was supported by the National Natural Science Foundation of China under Grant No.42504030 and No.52302506; the Fundamental Research Funds for the Central Universities (Science and Technology Program) under Grant No.D5000250047; and the Shaanxi Key Research and Development Program under Grant No.2025GH-YBXM-022.

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
