# OpenReview forum: "No Object Is an Island: Enhancing 3D Semantic Segmentation Generalization with Diffusion Models"
_NeurIPS.cc/2025/Conference — NeurIPS 2025 poster_

### Official Review · Reviewer_4ZFB · 2025-07-01

**Clarity:** 3
**Significance:** 3
**Originality:** 3
**Rating:** 4
**Confidence:** 4

**Summary:**

This paper studies 3D Semantic Segmentation Generalization enhanced by diffusion models. Specifically, it aims to harness the ability from diffusion models to capture underlying contextual distributions and spatial relationships among objects to tackle the challenging task of cross-domain 3D semantic segmentation. To achieve it, it designs  Dual-queries Consistency Scheme to extract features from diffusion model and construct Consistency Constraint between  queries from different timesteps. The experimental results show that the proposed method outperforms previous methods.

**Questions:**

- The authors claim that it uses diffusion models to address the dependency between objects. However, it does not provide evidence to support the motivation. It is recommended to provide detailed analysis (visualization and/or quantitative results).

- The experimental results show that the diffusion model indeed bring improvements. But the improvements really result from  "the diffusion model’s ability to flexibly compose diverse objects into high-quality images across varying domains"? Please provide evidence. In addition, can we use other visual foundation models to achieve it?

- It seems that the number of parameters of the proposed method is higher than previous methods due to the introduction of diffusion models. Thus, please provide the parameters for each method and provide a more fair comparison.

**Ethical Concerns:**

["NO or VERY MINOR ethics concerns only"]

**Final Justification:**

My concerns are addressed. After reading other reviews, I believe this paper could be accepted with slight modifications.

**Limitations:**

The authors note a limitation: 'the lack of source-domain data makes it reliant on paired images and point clouds in the target domain.' However, the reviewer is more interested in scenarios where the method underperforms and the reasons behind a limitation.

**Quality:**

3

**Strengths And Weaknesses:**

Strengths:

- The method is novelty and effective.

- The ablation can show the effectiveness of each components.

Weakness:

- The authors claim that it uses diffusion models to address the dependency between objects. However, it does not provide evidence to support the motivation. It is recommended to provide detailed analysis (visualization and/or quantitative results).

- The experimental results show that the diffusion model indeed bring improvements. But the improvements really result from  "the diffusion model’s ability to flexibly compose diverse objects into high-quality images across varying domains"? Please provide evidence. In addition, can we use other visual foundation models to achieve it?

- It seems that the number of parameters of the proposed method is higher than previous methods due to the introduction of diffusion models. Thus, please provide the parameters for each method and provide a more fair comparison.

---

> ### Author Rebuttal · Authors · 2025-07-31
>
> **Q1**: To substantiate the claim that diffusion models help capture inter-object dependencies, it is recommended to include detailed analysis through visualizations and/or quantitative results.
>
>   **A**: Thank you for the thoughtful comment. As illustrated in Figures 1 and 2 of the supplementary material (see lines 87–96 in Section A.5), our method shows clear improvements over the baseline (UniDSeg) in capturing semantic dependencies and spatial arrangements. Specifically, it exhibits more comprehensive spatial coverage of objects, contextually consistent category predictions, and effective suppression of fragmented outputs.
>
>   For instance, in Figure 1 (row 1), our method produces a more coherent segmentation of the "road" region, ensuring comprehensive spatial coverage. In Figure 1 (row 4), it avoids misclassifying "vegetation" as "truck", resulting in contextually consistent category predictions. Similarly, in Figure 2 (row 2), our method significantly reduces fragmented or implausible predictions for objects. To further demonstrate the model’s ability to capture semantic dependencies, we will include more comprehensive and systematic visual results, such as clustering-based visualizations of intermediate features, in the final version.
>
> **Q2**: It is unclear whether the performance gain from the diffusion model comes from its cross-domain compositional ability. Please provide evidence and clarify if other vision foundation models could achieve similar results.
>
>   **A**: Thank you for the thoughtful question. Figures 1 and 2 in the supplementary material intuitively demonstrate that our method achieves a deeper understanding of scene semantic distributions than the baseline (UniDSeg), as evidenced by broader spatial coverage of objects, contextually consistent predictions, and reduced fragmented outputs (see lines 87–96 in Section A.5). These improvements can be attributed to the scene-level priors of the diffusion model, due to its unique ability to compose diverse objects into coherent scenes across varying domains.
>
>   To further support this, we conducted ablation studies replacing diffusion features with those from vision foundation models (VFMs) designed for image understanding, such as CLIP and DINOv2. As shown in the table below, only the diffusion model consistently yielded significant performance gains, while the VFM-based variants underperformed. Even advanced multimodal models like Qwen2.5-VL—which exhibit stronger image-text understanding capabilities—showed inferior performance compared to diffusion model. This underscores the unique contribution of diffusion models in our framework.
>
>   Unlike VFMs that focus on capturing category-specific attributes of individual objects, diffusion models are trained to generate coherent and realistic images, requiring them to model spatial layouts, semantic relations, and contextual dependencies. These structured priors enable our method to better capture domain-invariant semantics, resulting in superior generalization. Thus, existing VFMs are unable to achieve comparable results to diffusion models.
>
>   | S:Source/T:Target | vKITTI/sKITTI | nuScenes:USA/Sing | nuScenes:Day/Night | A2D2/sKITT |
>   | ----------------- | ------------- | ----------------- | ------------------ | ---------- |
>   | CLIP              | 44.5          | 64.9              | 70.7               | 46.8       |
>   | DINO              | 44.8          | 64.1              | 69.8               | 47.3       |
>   | DINOv2            | 45.0          | 64.6              | 69.7               | 47.6       |
>   | Qwen2.5-VL        | 45.6          | 65.3              | 71.0               | 48.1       |
>   | Ours (Diffusion)  | 46.9          | 66.7              | 72.4               | 49.1       |
>
> **Q3**: The proposed method appears to have more parameters due to the diffusion model. Please provide parameter counts for each method to enable a fair comparison.
>
>   **A**: Thank you for the valuable comment. While our method incorporates a diffusion model during training, we would like to clarify that the diffusion model remains entirely frozen and is not used during inference. As a result, it introduces no additional trainable parameters and does not increase the model size at test time. The only additional trainable components are the agent queries, which introduce a relatively minor parameter overhead compared to the whole model. As shown in the table below, the total number of parameters in our method remains comparable to the baseline, with a minimal increase that does not affect inference efficiency.
>
>   | Method             | Trainable Params | Inference Params |
>   | ------------------ | ---------------- | ---------------- |
>   | UniDSeg (baseline) | 38.5M            | 38.5M            |
>   | Ours               | 40.3M            | 40.3M            |
>
> **Q4**: The noted limitation focuses on data dependency, but the reviewer is more interested in failure cases and their underlying causes.
>
>   **A**: Thank you for the valuable suggestion. In our experiments, we employed Stable Diffusion v2, which was trained on a relatively limited range of visual domains and semantic classes. This constraint made it easier to extract scene-level semantic priors. However, it also limits generalization when encountering out-of-distribution inputs, leading to potential failure cases in unfamiliar environments. While adopting more recent diffusion models—such as Flux or Stable Diffusion v3—could improve domain coverage, it comes with substantially higher computational cost. Moreover, these models are trained on extremely broad and diverse datasets, leading to highly entangled semantic spaces. As a result, it becomes considerably more challenging to identify suitable timesteps and to extract scene-level semantic priors, as these are often obscured by entangled low-level visual cues such as textures and stylized patterns.

---

> > ### Comment · Reviewer_4ZFB · 2025-08-08
> > **Thanks for response**
> >
> > Thank the authors for their response. My concerns are addressed.

---

### Official Review · Reviewer_nDRB · 2025-07-02

**Clarity:** 3
**Significance:** 3
**Originality:** 3
**Rating:** 4
**Confidence:** 3

**Summary:**

This paper introduces XDiff3D, a novel cross-modal learning framework based on diffusion models to enhance the cross-domain generalization of 3D semantic segmentation. The authors leverage diffusion models to capture semantic dependencies among objects, constructing object agent queries to infuse 2D diffusion priors into the 3D representation space. A dual-queries consistency scheme is proposed to suppress visual details and focus on domain-invariant semantics. Experiments across multiple benchmarks demonstrate state-of-the-art performance, outperforming existing methods in various settings.

**Questions:**

1. Computational Efficiency: Since the diffusion model is applied, it is neccessraty to provide a direct comparison of computational resources between XDiff3D and other methods (e.g., training time, memory usage, FLOPs).
2. Semantic Dependency Validation: The object agent queries claim to capture "semantic dependencies". Could attention maps or correlation metrics (e.g., inter-object affinity scores) be provided for the justification over other methods?
3. Impact of different diffusion layers: The author introduced the dual-queries consistency scheme to suppress visual details. Have the authors considered not utilizing all diffusion layers from the start? It would be beneficial to conduct experiments exploring the selection of diffusion layers to determine their impact on the model's performance and the suppression of visual details.
4. Failure case analysis in limitation: In the supplementary material, the authors outline certain limitations; however, the discussion primarily centers around input constraints. To provide a more comprehensive understanding, it is advisable to include specific failure cases encountered when applying the proposed methods, accompanied by in-depth analyses of the underlying causes. This approach would not only enhance transparency but also offer valuable insights for future improvements and applications.

**Ethical Concerns:**

["NO or VERY MINOR ethics concerns only"]

**Final Justification:**

I decided to keep my score because the proposed method demonstrates clear improvements over existing baselines, and most of my initial concerns have been addressed. While some limitations remain—particularly regarding generalization vs. semantic disentanglement—the authors have acknowledged these and committed to further visualizations and analysis in the final version.

**Limitations:**

Yes

**Paper Formatting Concerns:**

None. Adheres to NeurIPS 2025 guidelines.

**Quality:**

3

**Strengths And Weaknesses:**

Strengths
•	Original Approach: The use of diffusion models to capture object-level semantic dependencies for 3D segmentation is innovative, addressing a gap in existing methods that treat objects in isolation.
•	Comprehensive Evaluation: The paper evaluates XDiff3D across diverse benchmarks, including synthetic-to-real, real-to-real, and normal-to-adverse weather transitions, showcasing its robustness.
•	Ablation Studies: Detailed ablation experiments validate the effectiveness of key components, such as object agent queries, dual-queries consistency, and loss functions.
•	Clear Contributions: The framework is well-motivated, with a clear explanation of how diffusion priors address limitations in existing 3D segmentation approaches.

Weaknesses
•	Computational Overhead: The use of diffusion models may introduce additional computational complexity.
•	Dependency on Paired Data: The method relies on paired images and point clouds, which could limit applicability in scenarios where multi-modal data is scarce.
•	Claim Justification: The method asserts its capability to effectively capture "semantic dependencies", yet lacks both quantitative or qualitative evidence, such as attention maps, to substantiate this claim.

---

> ### Author Rebuttal · Authors · 2025-07-31
>
> **Q1**: Since the diffusion model is applied, it is necessary to provide a direct comparison of computational resources between XDiff3D and other methods (e.g., training time, memory usage, FLOPs).
>
>   **A**: Thank you for pointing this out. In our framework, the diffusion model is used only during training to provide semantic priors, resulting in a slight increase in training time and memory usage compared to the baseline, as shown in the table below. Notably, all diffusion-related processes are conducted entirely offline and can be further optimized through engineering techniques to reduce training overhead. In addition, the diffusion model is not involved during inference, so the overall computational complexity at test time remains unchanged.
>
>   | Method             | Training time | Memory usage | FLOPs |
>   | ------------------ | ------------- | ------------ | ----- |
>   | UniDSeg (baseline) | 26h           | 11.1G        | 190M  |
>   | Ours               | 28h           | 15.7G        | 190M  |
>
> **Q2**: The method relies on paired images and point clouds, which could limit applicability in scenarios where multi-modal data is scarce.
>
>   **A**: Thank you for your insightful comment. To clarify, our method does not require paired RGB images and point clouds in the target domain. In both unsupervised domain adaptation (UDA) and domain generalization (DG) settings, training is conducted using paired RGB–point cloud data only in the source domain. This design improves the practicality of our approach, as paired data is typically easier to obtain in the source domain (e.g., via simulation or controlled capture). We will highlight this point more clearly in the final version.
>
> **Q3**: The method claims to effectively capture "semantic dependencies", yet lacks quantitative or qualitative evidence to support this claim.
>
>   **A**: Thank you for the thoughtful comment. As illustrated in Figures 1 and 2 of the supplementary material (see lines 87–96 in Section A.5), our method shows clear improvements over the baseline (UniDSeg) in capturing semantic dependencies and spatial arrangements. Specifically, it exhibits more comprehensive spatial coverage of objects, contextually consistent category predictions, and effective suppression of fragmented outputs.
>
>   For instance, in Figure 1 (row 1), our method produces a more coherent segmentation of the "road" region, ensuring comprehensive spatial coverage. In Figure 1 (row 4), it avoids misclassifying "vegetation" as "truck", resulting in contextually consistent category predictions. Similarly, in Figure 2 (row 2), our method significantly reduces fragmented or implausible predictions for objects. To further demonstrate the model’s ability to capture semantic dependencies, we will include more comprehensive and systematic visual results, such as clustering-based visualizations of intermediate features, in the final version.
>
> **Q4**: It would be valuable to investigate whether selectively using diffusion layers, instead of all from the beginning, affects model performance and the suppression of visual details.
>
>   **A**: Thank you for the valuable suggestion. For fine-grained perception tasks such as 3D semantic segmentation, preserving low-level visual cues—especially at object boundaries—is crucial. Discarding shallower diffusion layers and relying only on high-level semantics from deeper layers can impair boundary localization and ultimately lead to suboptimal performance. To validate this, we conducted an ablation study using different subsets of diffusion layers. As shown in the table below, the model achieves optimal performance only when the full range of diffusion features—from both shallow and deep layers—is utilized. This underscores the importance of multi-level diffusion information, which integrates fine-grained edge details with global semantic context to enable accurate and robust segmentation.
>
>   | Layer                      | vKITTI/sKITTI | nuScenes:USA/Sing | nuScenes:Day/Night | A2D2/sKITT |
>   | -------------------------- | ------------- | ----------------- | ------------------ | ---------- |
>   | 12                         | 44.6          | 64.7              | 70.5               | 46.2       |
>   | 11,12                      | 44.6          | 65.0              | 70.7               | 46.4       |
>   | 10,11,12                   | 44.8          | 64.9              | 70.8               | 46.2       |
>   | 7,8,9,10,11,12             | 45.4          | 65.5              | 71.4               | 46.9       |
>   | 4,5,6,7,8,9,10,11,12       | 46.3          | 66.3              | 71.9               | 48.2       |
>   | 1,2,3,4,5,6,7,8,9,10,11,12 | 46.9          | 66.7              | 72.4               | 49.1       |
>
> **Q5**: While the supplementary material outlines input-related limitations, including concrete failure cases with analysis would offer deeper insight and support future enhancements.
>
>   **A**: Thank you for the valuable suggestion. In our experiments, we employed Stable Diffusion v2, which was trained on a relatively limited range of visual domains and semantic classes. This constraint made it easier to extract scene-level semantic priors. However, it also limits generalization when encountering out-of-distribution inputs, leading to potential failure cases in unfamiliar environments. While adopting more recent diffusion models—such as Flux or Stable Diffusion v3—could improve domain coverage, it comes with substantially higher computational cost. Moreover, these models are trained on extremely broad and diverse datasets, leading to highly entangled semantic spaces. As a result, it becomes considerably more challenging to identify suitable timesteps and to extract scene-level semantic priors, as these are often obscured by entangled low-level visual cues such as textures and stylized patterns.

---

> > ### Comment · Reviewer_nDRB · 2025-08-07
> >
> > Thank you for the detailed rebuttal and the efforts in conducting the experiments. Most of my concerns have been addressed. I understand that due to changes in the rebuttal policy (e.g., disallowing global PDF attachments), the authors were unable to provide intermediate visualizations that would effectively verify the diffusion-related processes. However, I appreciate the authors’ commitment to include these visualizations in the final version.
> > ﻿
> > However, other reviewers have also raised questions about failure cases; while I find the discussion on failure cases still somewhat abstract — focusing primarily on the generalization limitations of the selected model rather than providing concrete examples. The authors also acknowledge that replacing the model with more advanced alternatives such as Stable Diffusion v3 or Flux might improve generalization but would lead to a more entangled semantic space. This, in my view, suggests a potential ("maybe unsolvable") trade-off inherent in the proposed method between generalization capability and semantic disentanglement, which merits further investigation.
> > ﻿
> > Nevertheless, the proposed approach still outperforms other baseline methods. Based on the above, I have decided to keep my score and look forward to the promised visualizations and further analysis in the final version.

---

### Official Review · Reviewer_Qzhy · 2025-07-02

**Clarity:** 2
**Significance:** 3
**Originality:** 3
**Rating:** 5
**Confidence:** 3

**Summary:**

This paper presents XDiff3D, a novel cross-modal framework leveraging diffusion models to improve the cross-domain generalization of 3D semantic segmentation. By constructing object agent queries from diffusion features, decoupling irrelevant local details, and enhancing 3D representations via cross-modal semantic alignment, the method captures object-level semantic dependencies more effectively. Extensive experiments demonstrate state-of-the-art performance across diverse domain adaptation benchmarks.

**Questions:**

1. How does the method compare to other approaches in terms of training and inference time, given the computational demands of diffusion models?

2. Is it possible to use more informative textual conditions (e.g., class names or semantic prompts) instead of empty text to better exploit the semantic potential of diffusion models?

3. How robust is the method to noisy or incomplete point cloud data, especially in target domains where LiDAR quality may vary?

4. Given that the method relies on paired RGB images and point clouds, how sensitive is it to calibration or alignment errors between the two modalities?

**Ethical Concerns:**

["NO or VERY MINOR ethics concerns only"]

**Final Justification:**

The authors have addressed all of my concerns with detailed clarifications and additional experiments. Their responses demonstrate the method's practicality, robustness, and efficiency. I am satisfied with the revisions and support acceptance of the paper.

**Limitations:**

yes

**Paper Formatting Concerns:**

No formatting issues noticed.

**Quality:**

3

**Strengths And Weaknesses:**

Strengths:
1. The writing is clear and well-organized.
2. The method is novel and builds a compelling cross-modal framework using diffusion models.
3. Experiments demonstrate strong performance with thorough ablation studies.

Weaknesses:
1. The paper lacks a comparison of training and inference time with other methods, which is particularly important given the potential computational overhead introduced by diffusion models.
2. The method depends on access to paired RGB images and point clouds in the target domain, which may hinder its practicality in real-world applications where obtaining fully aligned multi-modal data is often challenging.
3. The method uses empty text as the textual condition, which may not fully activate the semantic potential of the diffusion model. In contrast, incorporating class names or prior semantic labels could yield more precise and informative semantic features.

---

> ### Author Rebuttal · Authors · 2025-07-31
>
> **Q1**: How does the method compare to other approaches in training and inference time, given the computational overhead of diffusion models?
>
>   **A**: Thank you for your valuable comment. We have added a comparison of both training and inference time with other methods. As shown in the table below, our method introduces a slight increase in training time due to the incorporation of diffusion models. However, since all diffusion-related computations are performed entirely offline, this overhead can be further reduced through practical engineering optimizations. Notably, the diffusion model is not involved during inference, and thus our method introduces no additional computational cost at test time compared to previous approaches.
>
>   | Method         | xMUDA | UniDSeg (baseline) | Ours |
>   | -------------- | ----- | ------------------ | ---- |
>   | Training time  | 20h   | 26h                | 28h  |
>   | Inference time | 16ms  | 19ms               | 19ms |
>
> **Q2**: The method relies on paired RGB images and point clouds in the target domain, which limits its practicality in real-world applications.
>
>   **A**: Thank you for your insightful comment. To clarify, our method does not require paired RGB images and point clouds in the target domain. In both unsupervised domain adaptation (UDA) and domain generalization (DG) settings, training is performed using paired RGB–point cloud data only in the source domain. This design improves the practicality of our approach, and we will highlight this point more clearly in the final version.
>
> **Q3**: Is it possible to use more informative textual conditions (e.g., class names or semantic prompts) instead of empty text to better exploit the semantic potential of diffusion models?
>
>   **A**: Thank you for the insightful suggestion. We experimented with simple prompts using class names but did not observe noticeable performance improvements. Notably, despite using empty prompts, our method already achieves competitive results compared to previous approaches, indicating that the diffusion model can still provide meaningful scene-level priors without explicit textual input. Incorporating more complex prompts would require more powerful multimodal models and significantly increase computational cost. The current design strikes a well-balanced trade-off between effectiveness and efficiency. Additionally, overly detailed prompts may introduce task-irrelevant information, potentially disrupting the extraction of scene-level semantic priors and degrading 3D segmentation performance.
>
> **Q4**: How robust is the method to noisy or incomplete point cloud data, especially in target domains where LiDAR quality may vary?
>
>   **A**: Thank you for the insightful question. To evaluate robustness to noisy point clouds, we conducted an ablation study on A2D2$\to$SemanticKITTI benchmark by injecting zero-mean Gaussian noise into the input point clouds, with the noise level controlled by the standard deviation $\sigma$. As shown in the table below, our method consistently outperforms the baseline across varying noise levels, demonstrating strong robustness.
>
>   In addition, we assessed the robustness of our method to noisy and incomplete point clouds via the Normal-to-Adverse generalization experiment (Table 2 of the paper). Prior studies [1, 2, 3] have shown that adverse weather degrades LiDAR signals by introducing spatial noise, reducing point density, and distorting reflection intensity, leading to noisy and incomplete point clouds that significantly challenge perception systems.
>
>   Despite these effects, our method consistently outperforms previous approaches under adverse conditions (Table 2 of the paper). We attribute this to the incorporation of scene-level semantic priors from the diffusion model, which enable the model to capture contextual relationships and inter-object semantic dependencies, effectively mitigating the impact of noisy or incomplete geometry. This is further supported by Figures 1 and 2 of the supplementary material, which demonstrate the robustness of our method to incomplete point clouds resulting from occlusions or limited viewpoints (e.g., Figure 2, row 8).
>
>   | $\sigma$ (m)       | 0.05 | 0.1  | 0.2  | 0.3  |
>   | ------------------ | ---- | ---- | ---- | ---- |
>   | UniDSeg (baseline) | 40.3 | 37.1 | 33.8 | 30.6 |
>   | Ours               | 44.7 | 43.2 | 39.6 | 34.7 |
>
>   [1] Zhao, Haimei, et al. "Unimix: Towards domain adaptive and generalizable lidar semantic segmentation in adverse weather." *Proceedings of the IEEE/CVF Conference on Computer Vision and Pattern Recognition*. 2024.
>
>   [2] Xiao, Aoran, et al. "3d semantic segmentation in the wild: Learning generalized models for adverse-condition point clouds." *Proceedings of the IEEE/CVF Conference on Computer Vision and Pattern Recognition*. 2023.
>
>   [3] Hahner, Martin, et al. "Fog simulation on real LiDAR point clouds for 3D object detection in adverse weather." *Proceedings of the IEEE/CVF international conference on computer vision*. 2021.
>
> **Q5**: Given that the method relies on paired RGB images and point clouds, how sensitive is it to calibration or alignment errors between the two modalities?
>
>   **A**: Thank you for the thoughtful question. Our method follows the commonly adopted standard setup in multi-modal semantic segmentation, which assumes well-calibrated and aligned RGB–point cloud pairs in the source domain. Accordingly, we do not account for calibration or alignment errors during training. Notably, in the target domain, our method does not require paired RGB images and point clouds, where only point clouds are needed. As a result, potential misalignment in the target domain does not affect the applicability or performance of our method.

---

> > ### Comment · Reviewer_Qzhy · 2025-08-05
> >
> > Thank you for the comprehensive and thoughtful rebuttal. I appreciate the authors’ efforts in providing detailed training and inference time comparisons, clarifications on data requirements, robustness evaluations. Based on the responses, I believe most of my concerns have been addressed, and I have decided to raise my score accordingly. I look forward to seeing the newly added robustness experiments included in the revised version.

---

### Official Review · Reviewer_gtR6 · 2025-07-13

**Clarity:** 2
**Significance:** 3
**Originality:** 3
**Rating:** 4
**Confidence:** 2

**Summary:**

This paper focuses on enhancing the cross-domain generalization of 3D semantic segmentation task. They propose a diffusion model based cross-modal learning framework named XDiff3D. XDiff3D proposes to leverage diffusion model priors for 3D semantic segmentation. It mainly includes three components: (1) Contruction of Object Agent Queries, (2) Dual-Queries Consistency Scheme and (3) Cross-Modal Semantic Enhancement.

**Questions:**

(1) Why aggregate the semantic information of objects using diffusion features to form the object agent queries? What are the advantages of using diffusion modes in module (a)?
(2) What's the effect of SVD in module (b)? Is SVD a standard method for feature quantization?

**Ethical Concerns:**

["NO or VERY MINOR ethics concerns only"]

**Final Justification:**

Thanks to the authors for comprehensive responses.

**Limitations:**

yes

**Paper Formatting Concerns:**

This paper does not have  major formatting issues.

**Quality:**

3

**Strengths And Weaknesses:**

Strengths
(1) XDiff3D propose to inject 2D diffusion priors into 3D representations via object agent queries to enhance cross-domain generalization.
(2) XDiff3D  achieves state-of-the-art results, and extensive experiments also show the effectiveness of the proposed method.

Weaknesses
(1) The framework in Fig.2 is complex, which makes it hard to follow. It's vital to show the core idea of the proposed method.

---

> ### Author Rebuttal · Authors · 2025-07-31
>
> **Q1**: Why aggregate the semantic information of objects using diffusion features to form the object agent queries? What are the advantages of using diffusion models in module (a)?
>
>   **A**: Thank you for the question. The motivation for aggregating semantic information from diffusion features to form object agent queries is discussed in lines 57–64 of the Introduction. In brief, the object agent queries act as an interface to inject scene-level priors from diffusion features into the segmentation representation space, while simultaneously suppressing low-level visual details that may introduce noise. This mechanism allows the model to exploit the rich scene semantic distribution encoded in diffusion features without being distracted by irrelevant textures, thereby enhancing generalization across domains.
>
>   The advantages of using diffusion models in module (a) are discussed in lines 38–45 of the Introduction. Briefly, diffusion models provide strong scene-level priors by capturing semantic relationships among objects. Leveraging these priors helps the model better learn semantic dependencies and spatial arrangements, ultimately improving generalization in 3D semantic segmentation.
>
> **Q2**: What's the effect of SVD in module (b)? Is SVD a standard method for feature quantization?
>
>   **A**: Thank you for the question. The effect of SVD in module (b) is explained in Section 3.4, lines 184–189. Inspired by SoMA [1], which shows that principal components derived from SVD in vision foundation models encode generalized semantic knowledge essential for strong generalization, we apply SVD to the object agent queries to emphasize domain-invariant semantics while suppressing low-level visual details, thereby enhancing cross-domain generalization.
>
>   While SVD is not a standard method for feature quantization, it is widely adopted for feature decomposition and compression [1,2,3]. By decomposing the feature space into high-rank and low-rank components,  SVD enables structured compression, mitigates overfitting, and strengthens the model’s capacity to extract robust and semantically meaningful representations.
>
>   [1] Yun, Seokju, et al. "SoMA: Singular Value Decomposed Minor Components Adaptation for Domain Generalizable Representation Learning." *Proceedings of the Computer Vision and Pattern Recognition Conference*. 2025.
>
>   [2] Sun, Yanpeng, et al. "Singular value fine-tuning: Few-shot segmentation requires few-parameters fine-tuning." *Advances in neural information processing systems* 35 (2022): 37484-37496.
>
>   [3] Wang, Yuan, Rui Sun, and Tianzhu Zhang. "Rethinking the correlation in few-shot segmentation: A buoys view." *Proceedings of the IEEE/CVF conference on computer vision and pattern recognition*. 2023.
>
> **Q3**: The framework in Fig.2 is complex, which makes it hard to follow.
>
>   **A**: Thank you for the helpful feedback. We will revise Figure 2 in the final version by adding sub-figures to clarify key components and highlight the core idea, making the framework more intuitive.

---

### Comment · Area_Chair_1ySp · 2025-08-03

Dear Reviewers,

Thanks for your hard work during the review process. We are now in the author-reviewer discussion period.

Please (1) carefully read all other reviews and the author responses; (2) start discussion with authors if you still have concerns as early as possible so that authors could have enough time to response; (3) acknowledge and update your final rating. Your engagement in the period is crucial for ACs to make the final recommendation.

Thanks,

AC

---

> ### Comment · Area_Chair_1ySp · 2025-08-05
>
> Dear Reviewers,
>
> As we're approaching the end of author-reviewer discussion period, please read the rebuttal and start discussion with the authors as soon as possible. If all your concerns have been addressed, please do tell them so. Please note that submitting mandatory acknowledgement without posting a single sentence to authors in discussions is not permitted. Please also note that __non-participating reviewers will receive possible penalties of this year's responsible reviewing initiative and future reviewing invitations.__
>
> Thanks,
>
> AC

---

### Note · Authors · 2025-08-14

We sincerely thank all reviewers for their thoughtful comments and positive recognition. We appreciate the comments highlighting the novelty of our idea (Reviewer Qzhy &  Reviewer nDRB & Reviewer 4ZFB), the clarity of our motivation (Reviewer nDRB), the well-organization of our writing (Reviewer Qzhy), as well as the thoroughness of our experiments and ablation studies (Reviewer gtR6 & Reviewer Qzhy & Reviewer nDRB & Reviewer 4ZFB).

This paper proposes the first diffusion-based cross-modal framework for domain-generalized 3D semantic segmentation, addressing a key gap in current methods that treat objects in isolation by injecting 2D diffusion priors into 3D representations via object-agent queries. We further propose a dual-queries scheme that suppresses extraneous low-level details to support robust 3D feature learning. Extensive experiments validate our approach’s effectiveness, and we will release the full code.

During the rebuttal, we carefully addressed all reviewer comments with detailed clarifications, additional experiments, and in-depth analyses. These included training/inference cost comparisons, clarification of target-domain data requirements, strengthened explanations of our ability to capture semantic dependencies, and robustness evaluations under noisy and incomplete point clouds. Three reviewers indicated that their concerns were resolved and raised no further questions, while one reviewer has not yet responded. We will incorporate all feedback in the final version, and we thank the reviewers once again for their constructive feedback, which has further improved the clarity and completeness of the paper.

As a final note, we sincerely thank Area Chair 1ySp for dedicated coordination and timely engagement, which greatly facilitated the resolution of concerns during the discussion phase.

---

### Decision · Program_Chairs · 2025-09-17

**Decision:**

Accept (poster)

**Comment:**

This paper presents a novel framework for improving the cross-domain generalization of 3D semantic segmentation through leveraging knowledge in pre-trained diffusion models. Reviewers acknowledged the contribution and strong performance of the proposed method, while initially raising some concerns, such as its efficiency and unsupported claims which require additional clarification.

After the rebuttal, the authors addressed most of the concerns, and all reviewers agreed to accept this paper. AC read all the reviews, author rebuttals, and the paper, and believes this is a good paper and recommends acceptance.